# Fe_2_O_3_ Nanoparticles Doped with Gd: Phase Transformations as a Result of Thermal Annealing

**DOI:** 10.3390/molecules26020457

**Published:** 2021-01-16

**Authors:** Artem Kozlovskiy, Kamila Egizbek, Maxim V. Zdorovets, Kayrat Kadyrzhanov

**Affiliations:** 1Engineering Profile Laboratory, L. N. Gumilyov Eurasian National University, Satpaev str. 5, Nur-Sultan 010008, Kazakhstan; kemelin@mail.ru (K.E.); mzdorovets@gmail.com (M.V.Z.); kayrat.kadyrzhanov@mail.ru (K.K.); 2Laboratory of Solid State Physics, The Institute of Nuclear Physics, Ibragimov str. 1, Almaty 050032, Kazakhstan; 3Department of Intelligent Information Technologies, Ural Federal University, Mira str. 19, 620002 Ekaterinburg, Russia

**Keywords:** magnetic nanoparticles, hyperthermia, phase transformations, mechanochemical mixing

## Abstract

The aim of this work is to study the effect of the phase composition of the synthesized Fe_2_O_3_-Gd_2_O_3_ nanoparticles on the efficiency of using magnetic hyperthermia as a basis for experiments. This class of structures is one of the most promising materials for biomedical applications and magnetic resonance imaging. In the course of the study, the dynamics of phase transformations of nanoparticles Fe_2_O_3_ → Fe_2_O_3_/GdFeO_3_ → GdFeO_3_ were established depending on the annealing temperature. It has been determined that the predominance of the GdFeO_3_ phase in the structure of nanoparticles leads to an increase in their size from 15 to 40 nm. However, during experiments to determine the resistance to degradation and corrosion, it was found that GdFeO_3_ nanoparticles have the highest corrosion resistance. During the hyperthermal tests, it was found that a change in the phase composition of nanoparticles, as well as their size, leads to an increase in the heating rate of nanoparticles, which can be further used for practical purposes.

## 1. Introduction

In the modern world, magnetic nanoparticles are one of the important classes of magnetic nanomaterials [1,2]. Their small size, large specific surface area, as well as crystal structure and resistance to degradation, open up broad prospects for application in various fields of science and technology. At the same time, there are significant differences in the magnetic properties of nanoparticles and bulk materials, since the giant magnetoresistance, values of coercivity, saturation magnetization and remanent magnetization make it possible to use nanoparticles as contrast agents for liquids, targeted drug delivery, water purification, cell separation, etc. [3,4,5,6]. Interest in magnetic nanoparticles, in particular, iron oxide nanoparticles, is due to their good biocompatibility, resistance to external influences, the possibility of surface modification, etc. [7,8,9,10].

Despite the wide range of applications of nanoparticles, as well as a large number of scientific works on this topic, interest in their study is not weakening [11,12,13,14,15]. At the same time, in recent years, more and more attention has been paid to research on the structural and phase modification of iron-containing nanoparticles, the main goal of which is to search for the possibility of increasing the efficiency of their application in biomedicine. Thus, one of the promising directions is the modification of iron oxide nanoparticles with gadolinium [16,17,18], which is a paramagnetic element with a large magnetic moment and a low rate of electronic relaxation. The most common applications for gadolinium-modified nanoparticles are MRI contrast agents or magnetic hyperthermia. Thus, for example, in a recent work by Rong Fu et al. [19], it is shown that the modification of iron oxide nanoparticles with gadolinium oxide and terbium is promising in reducing the time of hyperthermal heating of nanoparticles, as well as increasing the specific absorption. The work of Fenfen Li et al. [20] presents the results of a core-shell study of Fe_3_O_4_/Gd_2_O_3_ nanoparticles and their applicability as contrast agents for MRI. In most of the works devoted to methods for the preparation of Fe_3_O_4_/Gd_2_O_3_ nanoparticles, the main emphasis is on methods for obtaining structures of the “core-shell” type or the usual modification of the surface of magnetic nanoparticles with gadolinium or its oxide. However, in these cases, the phase composition as well as the magnetic properties of nanoparticles can be strongly distorted due to nonequilibrium phases in the structure or impurity inclusions. In addition, most of the synthesis methods imply a complex chain of sequential modifications and operations, which entails high energy and resource consumption. However, despite the large number of works in this area [8,15,16,17,18,19,20,21], studies related to the search for optimal conditions for obtaining nanoparticles and the effect of their structural and phase modifications on biocompatibility and toxicity are still relevant and have great prospects for study.

Based on the above, the aim of this work is to study the effect of the phase composition of the synthesized Fe_2_O_3_-Gd_2_O_3_ nanoparticles on the efficiency of using magnetic hyperthermia as a basis for experiments. As a method of obtaining Fe_2_O_3_-Gd_2_O_3_ nanoparticles, a combination of a two-stage method is proposed. The first stage of the synthesis included the chemical synthesis of Fe_3_O_4_ nanoparticles, which were further mixed with Gd(NO_3_)_3_ and subjected to thermal annealing at different temperatures. Annealing is necessary to initiate the processes of phase and structural transformations, as well as to order the crystal structure.

## 2. Experimental Part

### 2.1. Method for Obtaining Nanoparticles

The following chemical reagents were used for the synthesis: FeCl_3_·6H_2_O, Gd(NO_3_)_3_, Na_2_SO_3_ manufactured by Sigma Aldrich (Saint Louis, MO, USA). The chemical purity of the reagents, according to the passport data, was 99.95%.

First stage: The chemical synthesis of Fe_3_O_4_ nanoparticles was carried out by dissolving 3.25 g of FeCl_3_·6H_2_O in 100 mL of water with the addition of 5 mL of Na_2_SO_3_ (5%). The resulting mixture was mixed with 20 mL of ammonia in an argon atmosphere and subsequent heating at 70 °C for 30 min. Thereafter, the resulting mixture was heated to 90 °C with the addition of citric acid for 90 min. The resulting precipitate, after cooling for 24 h, is washed to remove chlorides [22,23].

Second stage: The synthesized Fe_3_O_4_ nanoparticles are mixed with Gd(NO_3_)_3_ in equal molar ratios and annealed for 5 h in the temperature range of 400–800 °C. The resulting nanoparticles under various annealing conditions were subsequently used for characterization and testing.

Determination of the size of nanoparticles was carried out by constructing dimensional diagrams based on transmission electron microscopy data, as well as by the laser diffraction analysis (LDA) method implemented on an ANALYSETTE 22 NanoTec plus device (semiconductor green laser λ = 532 nm) with an ultrasonic dispersion unit in a liquid medium.

### 2.2. Study of Phase and Structural Features

The study of phase transformations and changes in structural parameters as a result of thermal annealing was carried out using the powder diffraction method performed on a D8 Advance Eco X-ray diffractometer (Bruker, Karlsruhe, Germany). The determination of the phases and the assessment of their content were carried out by a full-profile analysis of the obtained diffractograms by the Rietveld method, implemented in the TOPAS v.4 program code.

The study of the morphological features of nanoparticles and the dynamics of their changes in the process of annealing was carried out using the method of transmission electron microscopy, implemented on a JEOL JEM—1400 Plus, acceleration voltage 120 kV.

### 2.3. Corrosion Tests

Modeling of the processes of degradation of nanoparticles under conditions as close as possible to a real environment was carried out using an isotonic Phosphate buffered saline (PBS) solution with pH = 7.4. The time interval of corrosion tests was 100 h, the solution temperature was 42 ± 1 °C. The construction of anamorphoses was carried out on the basis of the obtained data on changes in the elemental and phase composition of nanoparticles at certain intervals.

### 2.4. Study of the Hyperthermal Properties of Nanoparticles

The study of the Hyperthermal properties of the synthesized nanoparticles was carried out using a variable frequency induction heating unit SPG-10AB-II Induction heater manufactured by Shuangping Power Supply Technologies Company Ltd., (Shanghai, China). Measurement conditions: current 20 A, alternating magnetic field with an amplitude of 210 Oe and a frequency of 320 kHz. Heating was carried out using a 6 cm diameter heating coil consisting of 4 turns. The determination of the magnetic field strength was calculated based on the current strength and the coil geometry. The uniformity of heating was ensured by the fact that the sample flask was completely located in the heating coil. The choice of these magnetic field conditions is due to the need to minimize the occurrence of side effects of exposure to magnetic fields. A test concentration of 10 mg nanoparticles was placed in a 30 mL aqueous solution inside the magnetic coil and isolated to prevent heat exchange with the environment. Temperature control was carried out using fiber-optic thermometers placed in the liquid; these changes were read every 5 s. Before testing, the nanoparticle solutions were additionally subjected to dispersion in an ultrasonic bath for 30 min, the purpose of which was to disperse the nanoparticles in order to avoid the formation of agglomerates characteristic of magnetic nanoparticles.

## 3. Results and Discussion

### 3.1. Analysis of Structural and Morphological Features of Nanoparticles

Figure 1 shows X-ray diffraction patterns reflecting the dynamics of phase transformations as a result of thermal annealing of nanoparticles. As a comparison, the diffraction pattern of the initial nanoparticles before annealing is shown, which clearly demonstrates the highly disordered amorphous structure of nanoparticles, the deformation of which is caused by the presence of a mixture of two phases in the composition of Fe_2_O_3_ and Fe_3_O_4_. As is known, the precise determination of the Fe_2_O_3_ and Fe_3_O_4_ phases using the X-ray diffraction method is difficult due to the close similarity of the structural features. A more accurate determination of the phase composition of iron-containing nanoparticles is possible using the method of Mössbauer spectroscopy, which makes it possible to determine the values of hyperfine magnetic fields with high accuracy, which are different for these phases. A preliminary analysis of the initial iron-containing nanoparticles showed that in the case of the initial nanoparticles, the values of hyperfine magnetic fields are 480–485 kOe, which are characteristic of the magnetite phase, while for annealed samples, the magnitude of the hyperfine magnetic fields exceeds 510–512 kOe, which is characteristic of partially disordered phases of hematite Fe_2_O_3_. The values obtained are also in good agreement with the previously presented data on phase transformations in [22].

As can be seen from the presented data, in the case of thermal annealing at 400 °C, the main phase is the Fe_2_O_3_ phase with the crystal lattice parameter a = 8.3249 Å. The absence of peaks characteristic of phases containing gadolinium or gadolinium oxide may be due to the fact that during mechanical grinding, gadolinium remains in an amorphous state and does not have a clearly pronounced crystalline phase. In the case of annealing at temperatures of 600 °С, the diffraction pattern show the appearance of new peaks with angular positions 2θ = 24.19°, 33.18°, 40.89°, 49.31°, having a different shape and width than the peaks characteristic of Fe_2_O_3_, which indicates the appearance of a new phase in the structure of nanoparticles. An increase in the crystal lattice parameters for the Fe_2_O_3_ phase a = 8.3461 Å may be associated with the partial replacement of iron atoms with gadolinium atoms, which leads to the formation of a new phase. Applying the Rietveld method for a full-profile analysis of the position of diffraction peaks and their width [24], it was found that new peaks are characteristic of the orthorhombic phase of GdFeO_3_ (PDF#00-059-0665). In this case, an increase in the annealing temperature leads to ordering and dominance of the GdFeO_3_ phase in the structure of nanoparticles. The crystal lattice parameters for the GdFeO3 phase are а = 5.3342 Å, b = 5.5946 Å, c = 7.6071 Å.

Additionally, the formation of the GdFeO_3_ phase leads to an enlargement of the crystallite sizes determined using the Williamson–Hall method [25]. For nanoparticles annealed at 400 °C, the average crystallite size is 15–17 nm. For samples annealed at a temperature of 600 °C, the size is 20–23 nm, while for the analysis of the peaks characteristic of the GdFeO_3_ phase, the crystallite size was 33–35 nm. For samples annealed at 800 °C, the crystallite size was 35–40 nm. Figure 1b shows the changes in the FWHM values by the Williamson–Hall method, which reflect the change in the degree of structural changes, as well as grain sizes. In the case of nanoparticles annealed at a temperature of 400 °C, a large slope angle indicates a distortion of the crystal structure. In the case of nanoparticles annealed at a temperature of 600 °C, a change in the FWHM value is observed and is divided into two curves, which correspond to two phases Fe_2_O_3_ and GdFeO_3_. In the case of nanoparticles annealed at 800 °C, the slope of the curve indicates a small contribution of deformations and distortions in the structure of nanoparticles, which indicates an increase in the degree of structural ordering in nanoparticles.

The change in the shape and size of nanoparticles is also confirmed by the data obtained using the method of transmission electron microscopy, presented in Figure 2.

Changing the annealing conditions leads first to the ordering of nanoparticles and a change in their shape, and then to their enlargement at 800 °C. For nanoparticles obtained at an annealing temperature of 800 °C, the formation of inclusions characteristic of the residual Fe_2_O_3_ phase is observed in the structure, as well as a thin amorphous-like porous interlayer on the surface, the size of which is no more than 1 nm (see data in Figure 2).

Thus, during the study, the kinetics of phase transformations was determined depending on the annealing temperature, which can be represented by the following scheme (see Figure 3). According to the presented scheme, as well as the transmission electron microscopy data, the synthesized nanoparticles are a “core-shell” structure at an annealing temperature of 600 °C and single-phase spherical particles at a temperature of 800 °C.

Figure 4 presents comparative diagrams of nanoparticle sizes determined using various methods: transmission electron microscopy and static light scattering.

As can be seen from the data presented, the size diagrams obtained by different methods correlate with each other and have good agreement in the case of annealed samples at temperatures of 600–800 °C. An insignificant deviation of the particle sizes determined using the X-ray diffraction method in the case of annealed nanoparticles at a temperature of 400 °C may be due to an unformed structure, as well as a strong distortion of the structure.

### 3.2. Research on Corrosion Properties and Hyperthermia

One of the key conditions for the applicability of nanoparticles as targeted drug delivery or biomedical purposes is their resistance to degradation and the rate of corrosion in various media. Knowing the rate of oxidation and phase degradation allows prediction of the time and area of application, as well as assessment of the consequences that nanoparticles may have for the body. In this case, as a rule, for iron-containing nanostructures, degradation mechanisms are associated with oxidation processes and phase transformations, which can lead to a deterioration in both structural and magnetic properties. These changes are primarily associated with the appearance of regions of structural disordering, a change in valence, the breaking of chemical bonds, and the appearance of a large number of oxygen vacancies in the structure. In the case of the use of nanoparticles in biomedicine, the main conditions for use in liquid media are acidity and temperature of the medium. As a rule, the modeling of the organism’s environment is carried out using an isotonic PBS solution, which makes it possible to closely model the environment of the human organism with pH = 7.4. The PBS solution contains salts of sodium chloride, sodium hydrogen phosphate, potassium chloride and potassium dihydrogen phosphate. The temperature of the environment also plays a significant role in the degradation process. Thus, for example, in [26], it was shown that an increase in the temperature of the medium leads to a significant acceleration of the degradation processes of iron-containing nanoparticles. The maximum approximate temperature of the environment, typical for the hyperthermal application of nanoparticles, is 42–43 °C, which allows the killing of cancer cells without affecting healthy ones. Figure 5 shows the results of the anamorphosis kinetics of the synthesized nanoparticles degradation for 100 h in a PBS solution at 42 ± 1 °С. To compare the effect of the formation of a shell from gadolinium on the rate of degradation of nanoparticles, additional studies of the kinetics of degradation of iron-containing particles were carried out. Nanoparticles were obtained according to the proposed method, except for the second stage of coating with gadolinium, and were also annealed in a given temperature range under the same conditions as particles with gadolinium. A more detailed presentation of the effect of thermal annealing on phase transformations and changes in the structural parameters of Fe_3_O_4_/Fe_2_O_3_ nanoparticles, as well as the degree of resistance to degradation, is shown in [22,23]. The authors of these works have found that thermal annealing at temperatures above 400 °C leads to an increase in the degree of structural ordering, as well as to a change in the phase composition of nanoparticles, followed by the dominance of the Fe_2_O_3_ hematite phase in the structure at temperatures above 600 °C. In addition, an increase in the annealing temperature led not only to a change in the phase composition and ordering of the values of the hyperfine magnetic field, but also to an increase in the size of nanoparticles, followed by a change in their shape from spherical to hexahedral or rhombic.

The assessment of the degree of degradation was carried out by plotting anamorphous or kinetic curves of degradation, which are graphs of the dependence of changes in the elemental and phase composition over time while in the medium. Determination of the order of the kinetic curve of degradation makes it possible to determine the kinetics and rate of the oxidation and destruction reaction. The data for plotting anamorphosis were obtained by analyzing the changes in the elemental and phase composition of the studied nanoparticles over time. The phase composition was estimated using the X-ray diffraction method, by taking diffractograms after each cycle, to determine the contribution of impurity phases to the change in the phase composition. The identification of impurity phases was carried out using the PDF2 database and the Rietveld method, which makes it possible to determine the contributions of phase inclusions with high accuracy. As impurity phases, FeO and FeOH phases, characteristic of oxidation of iron, were considered. Dynamics of elemental ratio change were carried out using energy dispersion analysis method, by determination of Fe, Gd, O component concentration change and comparison of the obtained data with initial samples.

As can be seen from the presented data, a two-stage nature of the degradation of nanoparticles is observed. The first stage is in the range of 0–45 h and 0–60 h for nanoparticles annealed at temperatures of 400 °C and 600–800 °C, respectively. This stage is characterized by small changes in the elemental and phase composition of nanoparticles, and indicates a high degree of stability of nanoparticles to oxidation in comparison with nanoparticles without gadolinium. At the same time, in contrast to nanoparticles with gadolinium, for Fe_2_O_3_ nanoparticles, the processes of degradation and oxidation occur at significantly intensities, and the degradation rate increases by a factor of 3–4 compared to the rate of degradation of Fe_2_O_3_ + Gd_2_O_3_ particles under similar conditions. Moreover, in both cases, a decrease in the degradation rate is observed with an increase in the annealing temperature, which leads to a change in the phase composition of nanoparticles.

It should be noted that the main changes associated with corrosion are due to the appearance in the structure of oxide and hydroxide phases of iron, with a highly disordered structure, which is almost amorphous. The second stage is characterized by a more intense increase in structural and phase changes, which indicates an acceleration of the processes of degradation and oxidation of nanoparticles.

Figure 6 shows the data on the change in the mass of nanoparticles, as well as its loss as a result of degradation. Mass loss was calculated by determining the mass of nanoparticles depending on the residence time in the medium. The degradation experiment was carried out in several parallels. When a predetermined time interval was reached, the nanoparticles were captured by a magnet, and the solution was poured, after which the particles were dried and weighed using an analytical balance. The mass loss of nanoparticles during the experiment was determined from the change in mass before and after a certain time interval.

The general nature of changes in the dynamics of changes in the mass of nanoparticles can be divided into two stages. The first stage is characterized by a positive increase in the mass of nanoparticles, which is due to the formation of oxide inclusions on the surface of nanoparticles and the introduction of oxygen into the structure of nanoparticles. This stage is characterized by a low rate of degradation of nanoparticles. The second stage is typical for the loss of mass of nanoparticles, which is due to partial amorphization and degradation with subsequent destruction of nanoparticles or their fragmentation as a result of amorphization. A decrease in the mass loss, as well as the degradation rate for nanoparticles annealed at a temperature of 600–800 °C, is due to a change in the phase composition, as well as an increase in the degree of structural ordering as a result of annealing. The formation of the GdFeO_3_ phase leads to an increase in the degree of corrosion resistance, as well as to the processes of oxidation of nanoparticles in the medium. A similar situation is observed for unmodified nanoparticles, for which thermal annealing also leads to an increase in the degree of stability and a decrease in the rate of degradation.

In addition to resistance to degradation, an important property of the applicability of magnetic nanoparticles in biomedicine is the time required for processing in order to heat the nanoparticles to a temperature that will initiate the processes of local destruction of cancerous tumors using an alternating magnetic field. In this case, the time spent on heating plays an important role in determining the released thermal energy and the specific absorption coefficient.

Figure 7 shows the dynamics of changes in the heating rate of aqueous solutions of the studied nanoparticles with a volume of 10 mg/mL in an alternating magnetic field at a frequency of 320 kHz. As a comparison, the curve of an aqueous solution without nanoparticles is shown, which shows the absence of the effect of the action of an alternating current magnetic field on the aqueous solution and its heating.

The heating rate to a hyperthermia temperature of 42–45 °C, which is characteristic of local death of tumor tissues without damaging healthy tissues for the studied nanoparticles, was 0.4 °С/s, 0.71 °С/s and 1.05 °С/s for Fe_2_O_3_ doped Gd-400 °С, Fe_2_O_3_ doped Gd-600 °С and Fe_2_O_3_ doped Gd-800 °С, respectively. At the same time, the formation of the GdFeO_3_ phase in the structure of nanoparticles leads to a decrease in the solution temperature in comparison with the rest of the samples. The specific absorption rate (SAR) was calculated according to Formula (1) [27]:
(1)$$SAR=\frac{M}{m_{nanoparticles}}C_{water}\frac{\Delta T}{\Delta t},$$
where *M* is the mass of the solution, m_nanoparticles_ is the mass of nanoparticles, *С_water_* is the specific heat capacity of the solution, and ∆*T*/*t*∆ is the slope of the temperature curve. *SAR* data are presented in the diagram. The measurements were carried out at a magnetic field amplitude of 210 Oe and a frequency of 320 kHz. The results of the *SAR* change are shown in Figure 8а.

A rapid heating rate is associated with a change in the phase composition and the formation of the GdFeO_3_ phase. As is known from the literature, the formation of structures containing gadolinium leads to an increase in the heating rate, as well as an increase in the specific absorption. However, as it was found in the course of studies of phase and structural transformations, an increase in the annealing temperature leads to an increase in the particle size, which also affects the magnetic characteristics. According to the measurement data, the values of ultrafine magnetic parameters determined using the method of Mössbauer spectroscopy change from 480 kOe for the initial nanoparticles, which corresponds to a highly disordered mixture of Fe_3_O_4_/Fe_2_O_3_ phases, to 510–512 kOe for samples annealed at 800 °C. A change in the phase composition, as well as the ordering of the magnetic and structural properties, leads to a change in the behavior of the hyperthermal characteristics. At the same time the substitution of Gd atoms for Fe atoms at the sites of the crystal lattice with the subsequent formation of the GdFeO_3_ phase leads to a change in the energy barriers to the relaxation of magnetic moments and, hence, in the heating rate. At the same time, a change in the magnetic properties, as well as the size of particles, leads to their aggregation upon application of a magnetic field, which leads to a change in the dynamics and rate of heating of nanoparticles.

An important characteristic for determining the effectiveness of the use of nanoparticles in hyperthermia are the values of intrinsic loss power (ILP), which characterize the dissipated thermal power of magnetic nanoparticles, and also allow comparison of the results obtained in different laboratory conditions.

Intrinsic loss power were determined using Formula (2) [28]:
(2)$$ILP=SAR/H^{2}f,$$
where *H* is the applied field strength, *f—*Frequency. Figure 8b shows a diagram of the change in the value of intrinsic loss power depending on the type of nanoparticles.

As can be seen from the presented data, a change in the phase composition of the synthesized nanoparticles as a result of heat treatment leads to a significant increase in the SAR value, which, for nanoparticles with the GdFeO_3_ phase, exceeds by 2.6 times the analogous value for nanoparticles obtained at an annealing temperature of 400 °C. Thus, it was established that a change in the phase composition leads to an increase in the specific absorption value, which indicates a large amount of heat released per unit time in the local area. Moreover, an increase in the specific absorption value leads to an increase in the intrinsic loss power value. Accordingly, the obtained values of intrinsic loss power are in good agreement with the literature data for similar values of magnetic iron-containing nanoparticles used in hyperthermia [29,30,31]. In addition, the change in SAR and ILP values can also be associated with an increase in the size of nanoparticles upon thermal annealing and subsequent phase transformation, which was shown in [31].

## 4. Conclusions

The main results of this work are the new data obtained on the dynamics of phase transformations of nanoparticles Fe_2_O_3_ → Fe_2_O_3_/GdFeO_3_ → GdFeO_3_, which were established depending on the annealing temperature. The resulting dynamics make it possible to predict the possibility of obtaining nanoparticles with a given phase composition and geometry.

It has been determined that the predominance of the GdFeO_3_ phase in the structure of nanoparticles leads to an increase in their size from 15 to 40 nm. However, in the course of experiments to determine the resistance to degradation and corrosion, it was found that GdFeO_3_ nanoparticles have the highest corrosion resistance. The obtained results of corrosion resistance will make it possible in the future to predict the prospects and field of application of nanoparticles when interacting with aggressive media, as well as to determine the time frame of applicability until the stage of complete destruction. During the hyperthermal tests, it was found that a change in the phase composition of nanoparticles leads to an increase in the heating rate of nanoparticles.

Further research will focus on the in-depth study of the effect of the phase composition of nanoparticles on their magnetic properties and ultra-thin magnetic structure, as well as its resistance to external effects.

Based on the studies conducted, the following conclusion can be drawn about the possibility of practical application of synthesized nanoparticles in the biomedical field, not only as a basis for treatment using the method of magnetic hyperthermia, but also as a basis for magnetic carriers of drugs.

## Figures and Tables

**Figure 1 molecules-26-00457-f001:**
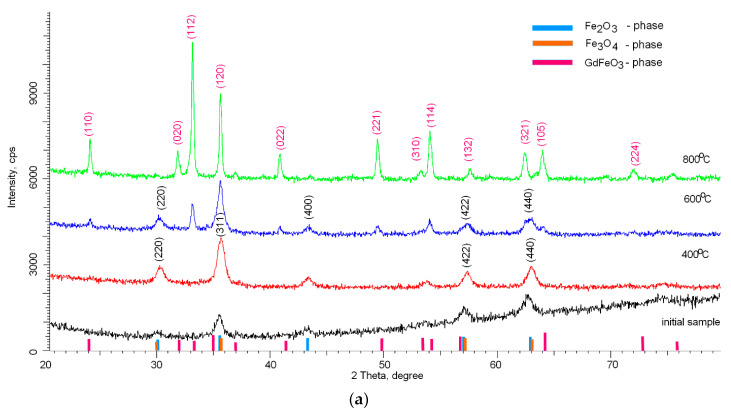
(**а**) X-ray diffraction patterns of synthesized nanoparticles versus annealing temperature (blue, orange and pink lines indicate the reference positions of the diffusion lines of the Fe_2_O_3_, Fe_3_O_4_, GdFeO_3_ phases, respectively, according to the PDF2 database); (**b**) plotting FWHM change data using the Williamson–Hall method.

**Figure 2 molecules-26-00457-f002:**
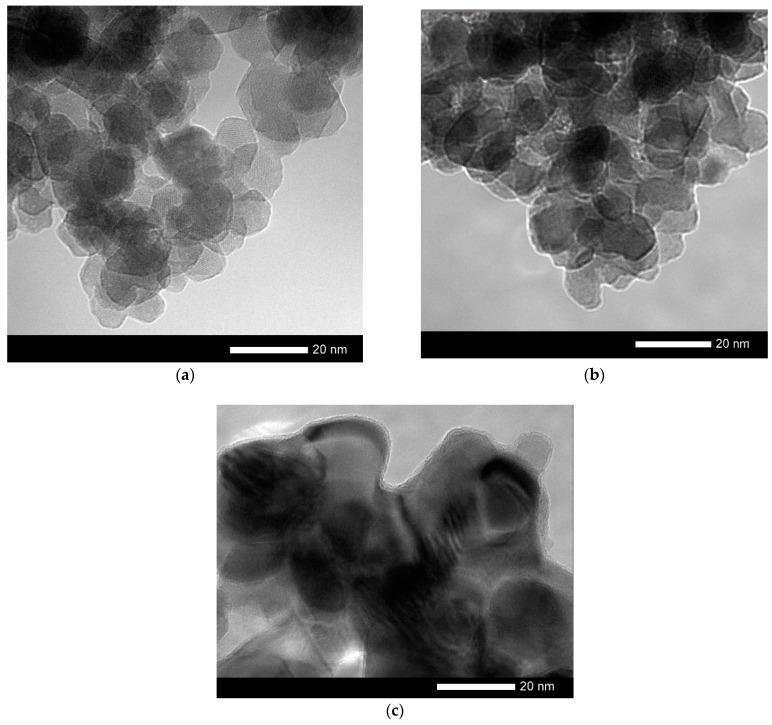
TEM—images of synthesized nanoparticles: (**a**) annealing at 400 °С; (**b**) annealing at 600 °С; (**c**) annealing at 800 °С.

**Figure 3 molecules-26-00457-f003:**
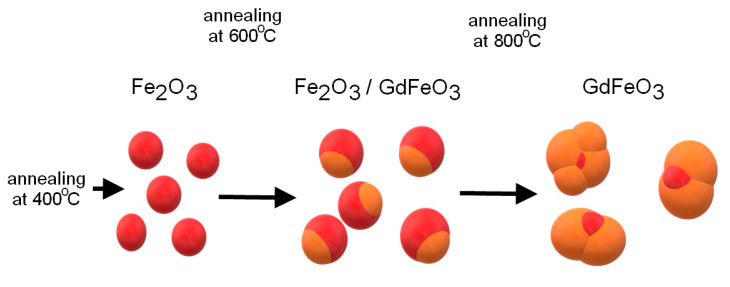
Schematic representation of phase transformations as a result of thermal annealing.

**Figure 4 molecules-26-00457-f004:**
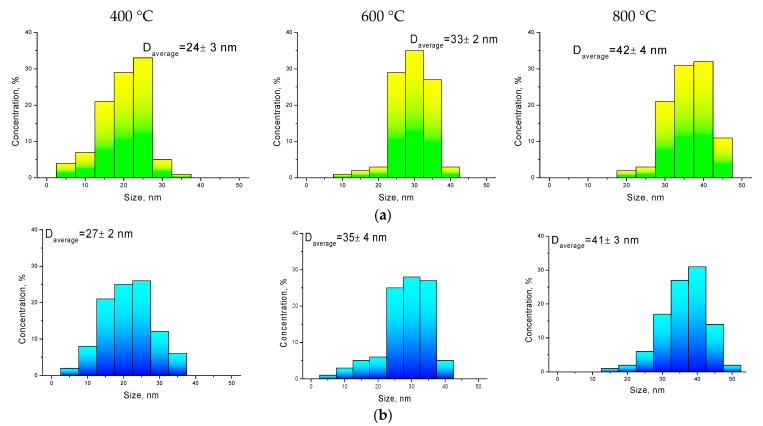
Calculated nanoparticle size data determined by (**a**) transmission electron microscopy and (**b**) static light scattering; (**c**) comparative diagram of nanoparticle sizes obtained by different methods.

**Figure 5 molecules-26-00457-f005:**
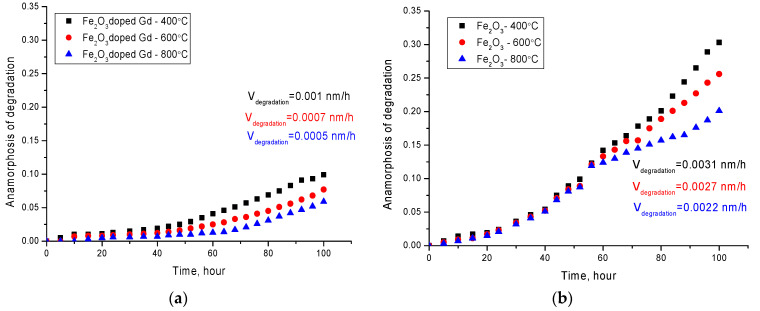
Dynamics of degradation anamorphosis of nanoparticles depending on the residence time in PBS at 42 ± 1 °С: (**a**) Fe_2_O_3_ + Gd_2_O_3_ nanoparticles; (**b**) Fe_2_O_3_ nanoparticles; (**c**) comparative degradation rate chart.

**Figure 6 molecules-26-00457-f006:**
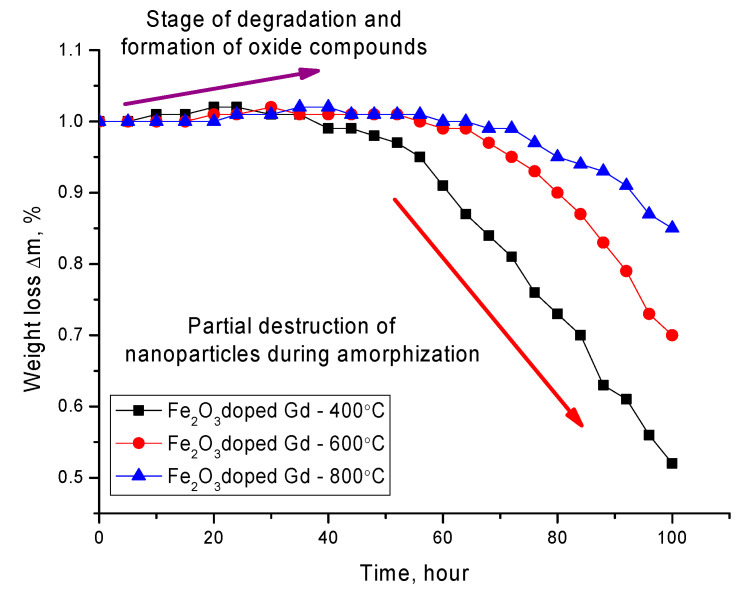
Graph of the dependence of the mass loss of nanoparticles as a result of degradation.

**Figure 7 molecules-26-00457-f007:**
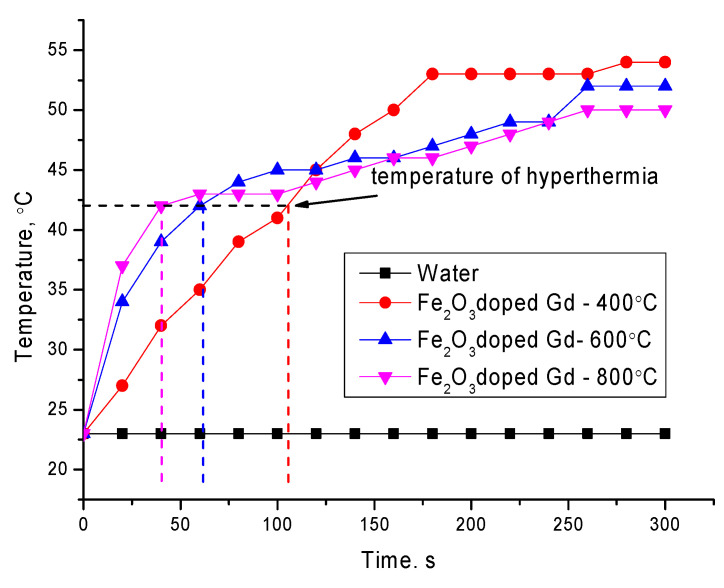
Graph of time dependence of heating of nanoparticles at a constant frequency of 320 kHz.

**Figure 8 molecules-26-00457-f008:**
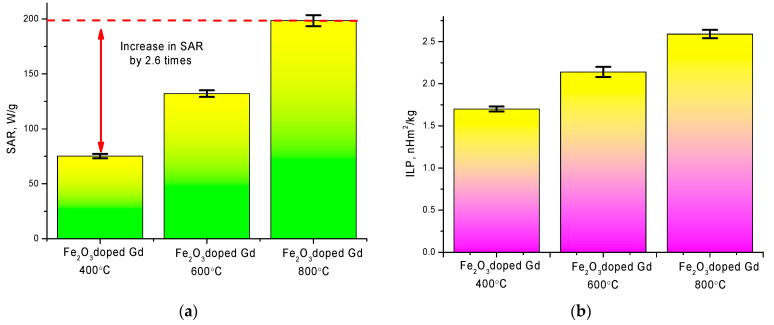
(**a**) Specific absorption rate (SAR) chart by type of nanoparticles; (**b**) intrinsic loss power value change diagram.

## Data Availability

Not applicable.

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
