# Peer review of "Fe2O3 Nanoparticles Doped with Gd: Phase Transformations as a Result of Thermal Annealing"

_molecules, 2021, doi:10.3390/molecules26020457_

Round 1

Reviewer 1 Report

This manuscript describes the preparation of different Gd-dopped hematite nanoparticles conducting to an increase in the oxidation stability and in the hyperthermia response of these nanoparticles.

In Figure 1 a and b, the colour of the legend doesn´t correspond to the colour of the graphs and one of the experimental results is missed in both legends.

FWHM values for nanoparticles prepared at 600 ºC annealed showed two different phases, one of hematite and other of Gd-dopped hematite. Does it affect to the hyperhtermia response?

TEM images showed agglomeration formation of the nanoparticles, especially the nanoparticles prepared at 800 ºC annealed. In fact, at 800 ºC annealing temperature it is observed that nanoparticles structure is not clear and they tend to unify in a whole mass. Authors should explain better the struture modifications and their consequences in the final applications.

Along the degradation studies, authors mentioned that a similar situation was observed for unmodified nanoparticles, for which thermal annealing also leads to an increase in the degree of stability and a decrease in the rate of degradation. However, these results are not shown in the manuscript.

Author Response

1. The authors thank the reviewer for this comment. Correction made.

Figure 1. а) X-ray diffraction patterns of synthesized nanoparticles versus annealing temperature (blue, orange and pink lines indicate the reference positions of the diffusion lines of the Fe2O3, Fe3O4, GdFeO3 phases, respectively, according to the PDF2 database);

2. Emergence of the second phase, in particular GdFeO3, leads to the fact that the heating rate of nanoparticles increases from 0.4 °Ð¡/s, 0.71 °Ð¡/s and 1.05 °Ð¡/s for Fe2O3 doped Gd-400°Ð¡, Fe2O3 doped Gd-600°Ð¡ and  Fe2O3 doped Gd-800°Ð¡, respectively. This is primarily due to a change in the phase composition, as well as the degree of structural ordering. The formation of the GdFeO3 phase at 600ºC and its subsequent dominance at an annealing temperature of 800ºC leads to an increase in the specific absorption value, as well as a large amount of heat released in the local volume. Also, the formation of the GdFeO3 phase in the structure of nanoparticles at annealing temperatures of 600-800°C leads to an increase in the size of nanoparticles, due to their adhesion and the formation of nonspherical particles, which are covered with an amorphous-like porous layer.

3. In the article, Figure 5 presents comparative data on changes in the rate of degradation for nanoparticles of various types.

A more detailed presentation of the effect of thermal annealing on phase transformations and changes in the structural parameters of Fe3O4/Fe2O3 nanoparticles, as well as the degree of resistance to degradation, is shown in [23,24]. The authors of these works have found that thermal annealing at temperatures above 400°C leads to an increase in the degree of structural ordering, as well as to a change in the phase composition of nanoparticles, followed by the dominance of the Fe2O3 hematite phase in the structure at temperatures above 600°C. Also, an increase in the annealing temperature led not only to a change in the phase composition and ordering of the values of the hyperfine magnetic field, but also to an increase in the size of nanoparticles, followed by a change in their shape from spherical to hexahedral or rhombic.

Reviewer 2 Report

The revised manuscript of the paper submitted is improved as compared to the first submission.

However, there are still some issues the authors must address before considering publication:

  1. The authors provide more details in the revised manuscript regarding the hyperthermia set-up. They claim that the magnetic field strength used in their device is 210 Oe. Please provide information on the method used to measure it, was calculated based on the electric current intensity and coil geometry ( number of turns, length) or was determined by using a pick-up coil
  2. Also, it is not clear if the magnetic field is the same all over the sample as the sample volume is not identified ( the author state that the sample volume is 10 mg/mL, but this is a concentration, not  a volume)
  3. In figure 1a please insert the sample for each diffraction pattern presented (it can be seen only for the “initial”) or describe it in the legend of the figure
  4. In figure 4 and the text, the authors refer to Dynamic Light Scattering (DLS). However, the device described in the Materials and methods chapter is based on Static Light Scattering. Probably this is the reason why the size measured by SLS is so close to the TEM size (in DLS one measures the hydrodynamic radius which is always larger than the TEM size).
  5. SAR and SLP  are different names for the same physical quantity, the heat released by a mass unit of magnetic nanoparticles in unit time. The authors should choose only one name (as in the literature both terms are used) and use it throughout all the paper. There is no need to give two separate formulas for SAR and SLP (eq.1 and eq.2). Eq.2 is the more accurate equation to be used, but because the mass and the specific heat capacity of the magnetic nanoparticles are much smaller as compared to eater they are neglected. The heat capacity of the tube and thermometer (A) could be used if it is significant. However, I do not understand why the authors mention the use of the alcoholic thermometer, as in the Materials and Method chapter they claim to use an optical fiber probe for measuring the temperature
  6. Please comment on the results presented in figure 7; why the initial heating rate is the highest for the sample annealed at 800 oC but the final temperature is lower for this sample; it is possible that the nanoparticles aggregate during the heating (nothing is mentioned about coating) and the largest nanoparticles with the largest magnetic moments are more prone to aggregate?;

Author Response

1-2. The authors thank the reviewer for this comment. Correction made.

Heating was carried out using a 6 cm diameter heating coil consisting of 4 turns. The uniformity of heating was ensured by the fact that the sample flask was completely located in the heating coil. The choice of these magnetic field conditions is due to the need to minimize the occurrence of side effects of exposure to magnetic fields. A test concentration of 10 mg nanoparticles was placed in a 30 ml aqueous solution inside the magnetic coil and isolated to prevent heat exchange with the environment.

3. The authors thank the reviewer for this comment. Correction made.

Figure 1. а) X-ray diffraction patterns of synthesized nanoparticles versus annealing temperature (blue, orange and pink lines indicate the reference positions of the diffusion lines of the Fe2O3, Fe3O4, GdFeO3 phases, respectively, according to the PDF2 database);

4. The authors express great gratitude to the reviewer for this clarification. Indeed, the method described in Materials and methods chapter is based on Static Light Scattering. Also, the text of the article has been corrected regarding the change in the description of the caption to Figure 4.

5. The authors thank the reviewer for this comment. Corresponding changes were made in the article concerning the removal of the equation for determining the specific loss power.

6. A rapid heating rate is associated with a change in the phase composition and the formation of the GdFeO3 phase. As is known from the literature, the formation of structures containing gadolinium leads to an increase in the heating rate, as well as an increase in the specific absorption. However, as it was found in the course of studies of phase and structural transformations, an increase in the annealing temperature leads to an increase in the particle size, which also affects the magnetic characteristics. According to the measurement data, the values of ultrafine magnetic parameters determined using the method of Mössbauer spectroscopy change from 480 kOe for the initial nanoparticles, which corresponds to a highly disordered mixture of Fe3O4/Fe2O3 phases, to 510-512 kOe for samples annealed at 800°C. A change in the phase composition, as well as the ordering of the magnetic and structural properties, leads to a change in the behavior of the hyperthermal characteristics. At the same time the substitution of Gd atoms for Fe atoms at the sites of the crystal lattice with the subsequent formation of the GdFeO3 phase leads to a change in the energy barriers to the relaxation of magnetic moments and, hence, in the heating rate. At the same time a change in the magnetic properties, as well as the size of particles, leads to their aggregation upon application of a magnetic field, which leads to a change in the dynamics and rate of heating of nanoparticles.

Reviewer 3 Report

Paper can be accepted after the following corrections:

  • Conclusions should be developed to highlight the most important progress presented in the paper. Please also highlight the practical applications.
  • Editorial quality of the figures is not suitable for the publication. Please carefully prepare figures before final editing.

Author Response

The authors thank the reviewer for this comment. Conclusion corrected.

The main results of this work are the new data obtained on the dynamics of phase transformations of nanoparticles Fe2O3 → Fe2O3/GdFeO3 → GdFeO3 was established depending on the annealing temperature. The resulting dynamics makes it possible to predict the possibility of obtaining nanoparticles with a given phase composition and geometry.

It has been determined that the predominance of the GdFeO3 phase in the structure of nanoparticles leads to an increase in their size from 15 to 40 nm. However, in the course of experiments to determine the resistance to degradation and corrosion, it was found that GdFeO3 nanoparticles have the highest corrosion resistance. The obtained results of corrosion resistance will make it possible in the future to predict the prospects and field of application of nanoparticles when interacting with aggressive media, as well as to determine the time frame of applicability until the stage of complete destruction. During the hyperthermal tests, it was found that a change in the phase composition of nanoparticles leads not only to an increase in the heating rate of nanoparticles, but also to an increase in the specific absorption value, which indicates a large amount of heat released per unit time in the local region.

Further research will focus on the in-depth study of the effect of the phase composition of nanoparticles on their magnetic properties and ultra-thin magnetic structure, as well as its resistance to external effects.

Based on the studies conducted the following conclusion can be drawn about the possibility of practical application of synthesized nanoparticles in the biomedical field, not only as a basis for treatment using the method of magnetic hyperthermia, but also as a basis for magnetic carriers of drugs.

The authors thank the reviewer for this comment. Changes have been made to the figures.

Round 2

Reviewer 2 Report

The authors answered most of the issues raised in the previous review.

However they did not indicate how they calculated/measured the magnetic field strength in the hyperthermia setup.

In the review of the first submission  we noticed that

the authors claim that the SAR values are complementary to the heating rates ("a change in the phase composition leads not only to an increase in the heating rate of nanoparticles but also to an increase in the specific absorption value"). In fact, the SAR values are calculated based on the heating rates (eq.1) so the two notions are directly connected."

The authors corrected the main manuscript according to this observation, but this sentence remained  in the paper's abstract and it should be corrected.

No further review is required provided the authors addressed the above issues.

Author Response

1. However they did not indicate how they calculated/measured the magnetic field strength in the hyperthermia setup.

  1. The authors thank the reviewer for this comment. Correction made.

"Heating was carried out using a 6 cm diameter heating coil consisting of 4 turns. The determination of the magnetic field strength was calculated based on the current strength and the coil geometry. The uniformity of heating was ensured by the fact that the sample flask was completely located in the heating coil."

2. In the review of the first submission  we noticed that

the authors claim that the SAR values are complementary to the heating rates ("a change in the phase composition leads not only to an increase in the heating rate of nanoparticles but also to an increase in the specific absorption value"). In fact, the SAR values are calculated based on the heating rates (eq.1) so the two notions are directly connected."

The authors corrected the main manuscript according to this observation, but this sentence remained  in the paper's abstract and it should be corrected.

2. The authors thank the reviewer for this comment. Correction made.

"During the hyperthermal tests, it was found that a change in the phase composition of nanoparticles, as well as their size, leads to an increase in the heating rate of nanoparticles, which can be further used for practical purposes."

This manuscript is a resubmission of an earlier submission. The following is a list of the peer review reports and author responses from that submission.

Round 1

Reviewer 1 Report

The paper is a well-organized and interesting study of the iron oxide nanoparticles (NPs) doped with Gd. It is quite an important contribution to the field and in my opinion, the paper can be published in Molecules after addressing the following points:

  1. Hyperthermia: The authors mentioned that such NPs can be used in magnetic fluid hyperthermia and they also demonstrated the heating abilities. First, the annealing procedure yield powders, which must be redispersed for the hyperthermia treatment. Consequently, the size of the primary aggregates is rather large comparing to the original NPs (some idea?). This fact should be clearly mentioned in the Introduction and also in the Results. Some technical issues: The amplitude of the HF magnetic field (MF) is not given. What model (single exponential, two-exponential, or other) was used to fit the data and how it matches the data (pls add a plot). Please, give also the ILP values as they are MF-dependent.
  2. Particle size: The particle size changed based on the XRD data (a decrease of the FWHM upon annealing). Please show the WH plots. In general the larger crystalline particle size (hence the magnetic size), the larger the SAR (see, e.g. Cannas et al, https://doi.org/10.1039/D0NA00134A). This should be properly discussed in the manuscript.

I suggest minor revision as adding some more information and interpretation is needed. In general, the paper is already in very good shape.

Reviewer 2 Report

The aim of this study is the preparation of Fe2O3-Gd2O3 nanoparticles  to be used as hyperthermia agents for biomedical applications. However, authors don´t describe the power of the magnetic field applied to carry out the experiments. It is very important to see if these nanoparticles are safe to be used in biomedical applications. Also, SAR values depend on the power of the magnetic field applied, so it is a very important parameter to be defined. Besides, intrinsic loss power (ILP) should be calculated, since it is an independent parameter from the frequency and power used and it is suitable to compare with other studies.

The hyperthermia measurements should be deeply define, for example equipment used, number of samples used (repetitions, standard desviation), type of temperature sensor or probe, etc.

In the introduction, authors referenced other research works that prepare similar nanoparticles with core-shell structure. Authors mentioned that the phase composition of these nanoparticles can be strongly distorted due to the core-shell structure. However, they don´t specify the structure or morphology of the nanoparticles prepared in this manuscript. 

TEM images show high tendency to form agglomerations, so a dynamic light scattering analysis is convenient in order to determine the particle size distribution of the nanoparticles obtained.

Figure 4 shows the dynamics of anamorphosis of degradation of the nanoparticles, but authors don´t explain how these measurements were carried out. Please explain the technique used.

The abstract describes in line 21 that GdFeO3 nanoparticles have the highest corrosion resistance, versus what? They should define the highest corrosion resistance compared to the rest of nanoparticles synthesized.

Line 34: two etc. are writen. Correct it.

Reviewer 3 Report

In the paper presented for review, the authors propose an interesting class of magnetic nanoparticles composed of Fe and Gd oxides. While core-shell nanostructures of Fe3O4@Gd2O3 were previously reported as contrast agents in MRI, in the present paper the phase transformations induced by thermal annealing of magnetite nanoparticles mixed with Gd compounds at high temperatures (400-800 0C) and their effects on hyperthermic properties of the magnetic nanoparticles are investigated.

The topic might be interesting but the paper lacks many experimental details so that the experiments cannot be reproduced by an independent researcher.

Please find below some observations:

  1. The source and purity of the reagents are not provided.
  2. It is claimed that the nanostructures were obtained by mechanochemical mixing chemically synthesized Fe3O4 nanoparticles with “commercial” Gd(NO3)3 (?)nanoparticles.
  3. In the experimental part, it is mentioned that the Fe3O4 nanoparticles were mixed with the gadolinium salt and annealed for 5 hours at different temperatures. It is not clear if the magnetite nanoparticles were mixed with gadolinium nanoparticles or with Gd nitrate salt. Later on, when the authors mention the gadolinium nanoparticles, they refer to Gd2O3.
  4. It is claimed that the corrosion test was carried out based on the obtained data on changes in the elemental and phase composition of nanoparticles at certain intervals. However, no details about the methods used for elemental and phase composition are provided.
  5. No details about the hyperthermia setup are provided (device, coil type, magnetic field strength, sample positioning, temperature recording, etc.)
  6. The main experimental data based on which the authors interpret the phase transformations are the X-ray diffractograms. In figure 1 the reflection plane indexes for each diffraction peak should be added. Eventually, the initial Fe3O4 nanoparticles’ diffractogram should be added to be compared to the 400oC annealed sample (useful to prove the phase transformation of magnetite in maghemite or hematite).
  7.  Size histograms derived from the TEM images should support the claimed size increase by increasing the annealing temperature. However, in figure 2 c it is not clear what is the size of an individual nanoparticle. 
  8. What is more exactly the "Anamorphosis of degradation" on the Y-axis in figure 4? The terminology is not quite common and might be misleading.
  9. How was measured the mass loss in figure 5? Why the mass loss is less pronounced as the annealing temperature increases?
  10. The heating rate should be measured in 0C/s (lines 192-193).
  11. The authors claim that the SAR values are complementary to the heating rates ("a change in the phase composition leads not only to an increase in the heating rate of nanoparticles but also to an increase in the specific absorption value"). In fact, the SAR values are calculated based on the heating rates (eq.1) so the two notions are directly connected. The SAR values are meaningless without providing the magnetic field strengths used in the experiment.